# Organocatalytic desymmetrization provides access to planar chiral [2.2]paracyclophanes

**Vojtěch Dočekal** [1] ✉, **Filip Koucký** [2], **Ivana Císařová**[2] **& Jan Veselý** [1]✉

Planar chiral [2.2]paracyclophanes consist of two functionalized benzene rings connected by two ethylene bridges. These organic compounds have a wide range of applications in asymmetric synthesis, as both ligands and catalysts, and in materials science, as polymers, energy materials and dyes. However, these molecules can only be accessed by enantiomer separation via (a) time-consuming chiral separations and (b) kinetic resolution approaches, often with a limited substrate scope, yielding both enantiomers. Here, we report a simple, efficient, metal-free protocol for organocatalytic desymmetrization of pro-chiral diformyl[2.2]paracyclophanes. Our detailed experimental mechanistic study highlights differences in the origin of enantiocontrol of *pseudo-para* and *pseudo-gem* diformyl derivatives in NHC catalyzed desymmetrizations based on whether a key Breslow intermediate is irreversibly or reversibly formed in this process. This gram-scale reaction enables a wide range of follow-up derivatizations of carbonyl groups, producing various enantiomerically pure planar chiral [2.2]paracyclophane derivatives, thereby underscoring the potential of this method.

Asymmetric organocatalysis uses small organic molecules as chiral catalysts to mimic biocatalytic processes, thereby expanding the chemical space[1-4]. Organocatalytic approaches are valuable tools for preparing enantiomerically pure compounds given the operational simplicity of their reactions, which frequently include water and air tolerance. In addition, commonly used organocatalysts are available in both enantiomeric forms, and often derived from natural sources, such as amino acids and alkaloids[5]. Yet, despite the diversity of organocatalysts, organocatalytic approaches have been focused on the preparation of chiral molecules containing central and axial chirality. Consequently, asymmetric organocatalysis applications remain overlooked, especially in the production of planar chiral molecules, such as [2.2]paracyclophane derivatives[6].

In [2.2]paracyclophanes, two benzene rings are covalently bound by two ethylene bridges at arene *para* positions. This molecular architecture suppresses the rotation of the benzene rings, providing [2.2]paracyclophanes with high configuration stability (up to 200 °C)[7] and planar chirality upon arene derivatization[8]. In fact, the first planar

chiral derivative of these compounds was isolated by crystallization of brucine salts of 4-carboxy[2.2]paracyclophane[9] only 6 years after Brown and Farthing had pioneered the preparation of [2.2] paracyclophane[10]. Since then, considerable research efforts have focused on the unique 3D structure of chiral [2.2]paracyclophanes for their unusual electronical[11,12] and photophysical properties[13-19]. Case in point, highly rigid planar chiral [2.2]paracyclophanes (Fig. 1) have become a valuable toolbox for developing ligands[20-24] and organocatalysts[25]. Beyond synthetic chemistry, these scaffolds have also been applied in small-organic circularly polarized luminescence (CPL, Fig. 1D)[26-28] and other phosphorescent emitters[29].

Notwithstanding these applications, enantiopure [2.2]paracyclophanes lack general and efficient synthetic pathways, a major constraint that continues to stall progress in this research field. Currently available synthetic approaches rely on enantiomer separations or various resolutions, including chemical resolution through diastereomerization and kinetic resolution[30]. Kinetic resolution, in particular, involves metal[31-35] and enzyme-catalyzed processes[36-38] and organocatalytic

[1]Department of Organic Chemistry, Faculty of Science, Charles University, Hlavova 2030/8, 128 43, Prague 2, Czech Republic. [2]Department of Inorganic Chemistry, Faculty of Science, Charles University, Hlavova 2030/8, 128 43, Prague 2, Czech Republic. ✉e-mail: vojtech.docekal@natur.cuni.cz; jan.vesely@natur.cuni.cz

methods[39–42] although the last approaches remain incipient. Regardless of the approach, though, kinetic resolution entails an inherent limitation, that is, the maximum enantipure product yield is only 50%. For a high-yielding and practical synthesis of chiral [2.2]paracyclophanes, desymmetrization or dynamic kinetic resolution can be used, but only one study has reported such an approach thus far, more specifically the desymmetrization of centrosymmetric diformyl[2.2]paracyclophanes by ruthenium-catalyzed asymmetric transfer hydrogenation[43]. Moreover, this method still has some limitations, not least of which a limited reaction scope. Therefore, facilitating synthethic access to enantipure [2.2]paracyclophanes requires developing high-yielding methods with a wide substrate scope.

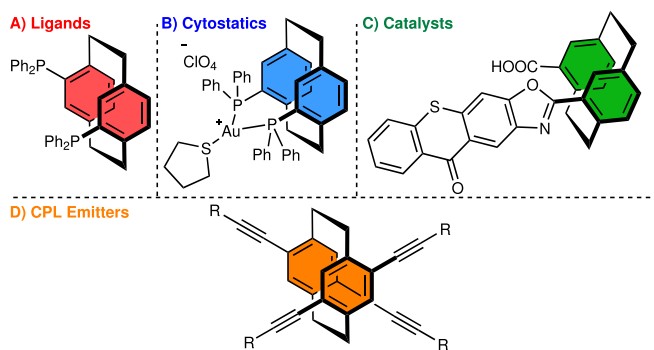

**Fig. 1 | Selected examples of chiral [2.2]paracyclophanes. A** Ligands (highlighted in red). **B** Cytostatics (highlighted in blue). **C** Catalyst (highlighted in green). **D** CPL Emitters (highlighted in orange).

Applicable to a broad scope of *meso*-symmetric substrates, metal-free organocatalytic desymmetrization[44–47] induced by chiral *N*-heterocyclic carbenes (NHCs) yields enantiomerically pure compounds[48–51]. Furthermore, NHC organocatalysis features versatile reactivity modes under mild reaction conditions, broad functional-group tolerance, and bench-stable NHC precursors derived from natural sources (such as amino acids). For example, oxidative NHC catalysis was applied to the atroposelective desymmetrization of aromatic dialdehyde, producing axially chiral monoesters[52,53]. However, NHC-catalyzed desymmetrization to planar chiral [2.2]paracyclophanes has never been attempted before. Nevertheless, a recent study has shown that NHC facilitates access to planar chiral ferrocenes via enantioselective desymmetrization[54]. Accordingly, we aimed at developing a method for preparing enantiomerically pure [2.2]paracyclophane derivates using an oxidative NHC-catalyzed process.

In this study, we report a highly efficient and versatile protocol for organocatalytic desymmetrization esterification of prochiral diformyl[2.2]paracyclophanes through NHC catalysis under mild conditions. For this purpose, we used amino acid-derived precursors to induce enantiocontrol via central-to-planar chirality transfer. After optimizing the reaction conditions, we analysed the reaction scope and conducted mechanistic studies to understand differences in the origin of enantiocontrol of organocatalytic desymmetrization.

## Results

### Optimization of reaction conditions

From the outset of our study, we chose the *pseudo-para* derivative (**1a**) as a model substrate considering the accessibility of prochiral diformyl[2.2]paracyclophanes. Simply mixing achiral paracyclophane **1a**

## Table 1 | Optimization studies of desymmetrization

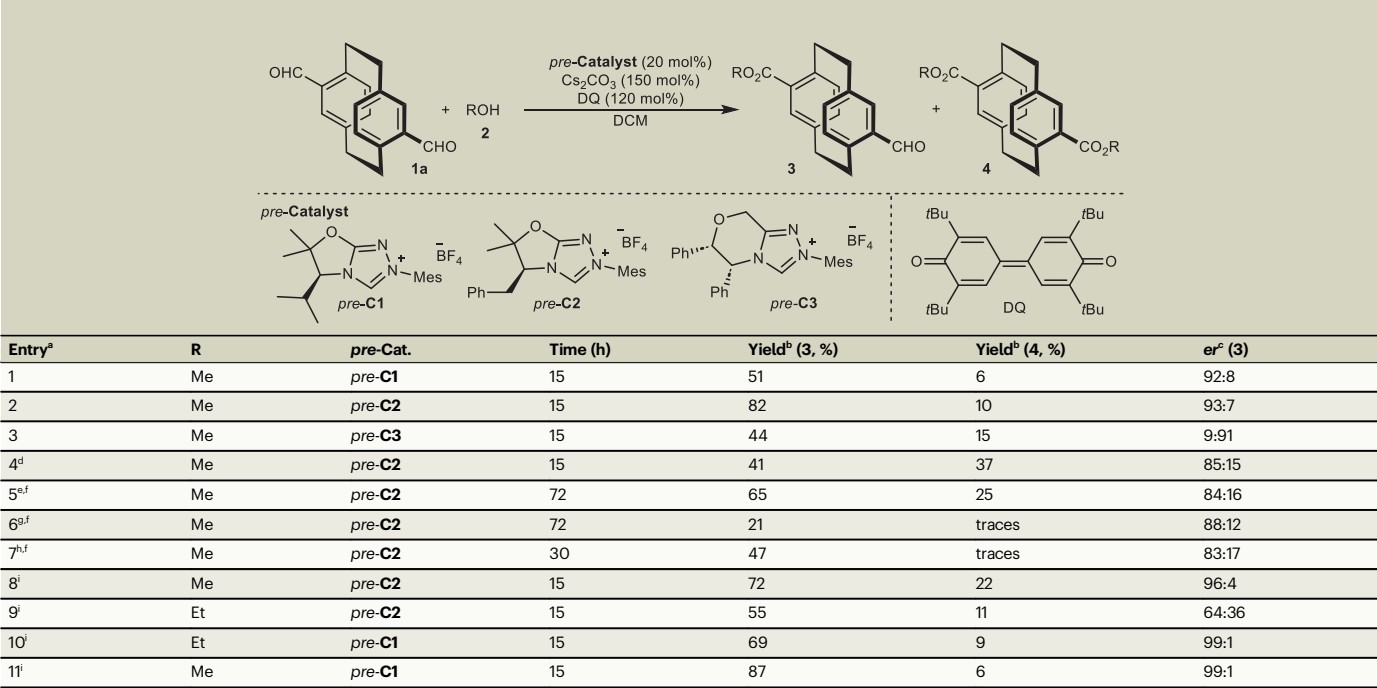

| Entry[a] | R | *pre*-Cat. | Time (h) | Yield[b] (3, %) | Yield[b] (4, %) | er[c] (3) |
|---|---|---|---|---|---|---|
| 1 | Me | *pre*-C1 | 15 | 51 | 6 | 92:8 |
| 2 | Me | *pre*-C2 | 15 | 82 | 10 | 93:7 |
| 3 | Me | *pre*-C3 | 15 | 44 | 15 | 9:91 |
| 4[d] | Me | *pre*-C2 | 15 | 41 | 37 | 85:15 |
| 5[e,f] | Me | *pre*-C2 | 72 | 65 | 25 | 84:16 |
| 6[g,f] | Me | *pre*-C2 | 72 | 21 | traces | 88:12 |
| 7[h,f] | Me | *pre*-C2 | 30 | 47 | traces | 83:17 |
| 8[i] | Me | *pre*-C2 | 15 | 72 | 22 | 96:4 |
| 9[i] | Et | *pre*-C2 | 15 | 55 | 11 | 64:36 |
| 10[i] | Et | *pre*-C1 | 15 | 69 | 9 | 99:1 |
| 11[i] | Me | *pre*-C1 | 15 | 87 | 6 | 99:1 |

[a] Reactions were conducted with **1a** (0.10 mmol), the corresponding alcohol **2** (0.5 mmol), Cs₂CO₃ (0.15 mmol), DQ (0.12 mmol), and *pre*-catalyst (20 mol%) in DCM (1.0 ml) at room temperature.
[b] Isolated yield after column chromatography.
[c] Determined by chiral HPLC analysis.
[d] CHCl₃ was used as a solvent.
[e] TEA was used as a base.
[f] Full consumption of **1a** was not observed.
[g] TEMPO was used as an oxidant.
[h] Electrochemical oxidation (Pt cathode and anode, constant current: 1 mA, total charge: 5.44 F/mol) using TBAI (0.2 mmol) in IKA ElectraSyn 2.0 was applied instead of DQ.
[i] 0.2 mmol of Cs₂CO₃ was used instead of 0.15 mmol. *Er*—enantiomeric ratio.

with an excess of methanol and in the presence of an L-valine-derived NHC-precursor (*pre*-**C1**), an oxidant (Kharash reagent, 3,3′5,5′-tetra-*tert*-butyldiphenoquinone, DQ), and a base (cesium carbonate) produced planar chiral monoester **3a** in 51% isolated yield with enantioselectivity 92:8 *er*, along with an easily separable diesterification by-

product (Table 1, entry 1). Based on the results from this proof-of-concept experiment, we aimed at optimizing the efficiency and stereochemical outcomes by varying the reaction conditions. For this purpose, we tested different amino acid-derived and other NHC precursors, oxidants, bases, and solvents.

**Fig. 2 | Reaction scope of the *pseudo-para* derivative. A** Scope of aliphatic alcohols (highlighted in red). **B** Scope of 2-arylethanols and related aromatic alcohols (highlighted in blue). **C** Scope of alcohols derived from natural or bioactive compounds (highlighted in green). **D** Scope of thiols (highlighted in orange).

The isolated yield of **3a** significantly increased in the model reaction (entry 2) mediated by an L-phenylalanine-derived NHC precursor (*pre*-**C2**). Other precursors, such as morpholine-based *pre*-**C3**, failed to improve the efficiency of this reaction. In addition to these amino acid-derived NHC precursors, we also tested various other NHC precursors (for further information on the optimization survey, please refer to the Supplementary Tables 1–9). As a result, the model reaction became less tolerant to bases and solvents. For instance, with triethylamine as a base or chloroform as a solvent, the model reaction displayed lower yield and enantiocontrol (entries 4, 5). The same outcome was found when replacing DQ by the single-electron oxidant TEMPO (entry 6). Conversely, electroredox oxidation using iodide as a promoter[55] produced the expected product **3a** in 47% yield, albeit slightly decreasing the enantiocontrol (entry 7). Nevertheless, this experiment validated electrochemical oxidation as a potentially more suitable approach than other systems involving additional oxidants.

After further optimizing the reaction conditions, we found that increasing the amount of base (2.0 equiv., entry 8) slightly improved the stereocontrol of the model reaction. Under optimized reaction conditions, we tested the desymmetrization approach using ethanol instead of methanol, but the enantiocontrol decreased significantly (entry 9). This decrease led us to reexamine the catalyst for esterification using ethanol. Surprisingly, the reaction mediated by *pre*-**C1** produced nearly an enantiopure product with a good yield (entry 10). Moreover, this reaction proved equally effective with methanol,

providing the desired product **3a** in excellent yield and enantiocontrol (entry 11).

## Reaction scope

After optimizing the reaction conditions, we began exploring the scope of the desymmetrization reaction of *pseudo-para* derivative **1a** (Fig. 2). When conducted with *ent*-*pre*-**C1** derived from unnatural D-valine, the desymmetrization reaction produced the expected opposite enantiomeric product (*ent*-**3a**) in high yield, albeit with slightly diminished enantiopurity. Then, we assessed the effect of the steric hindrance of the selected aliphatic alcohols on the reaction rate and stereochemical outcome (Fig. 2A). Unsurprisingly, the reaction rate was significantly slower when using sterically hindered alcohols. Conversely, longer aliphatic alcohols, such as lauryl alcohol, produced the corresponding ester **3d** in high yield (87%) and enantiopurity (94:6 *er*). Substituted aliphatic alcohols with halogen, methoxy, or internal and terminal alkenyl or alkynyl groups showed similar efficiency.

Subsequently, we explored the scope of this method using various aromatic alcohols (Fig. 2B). The results showed that this method was intolerant to phenols, including substituted phenols, but tolerated well benzyl alcohol and 2-phenylethanol. In addition, the expected products (**3m** and **3n**) were formed in high yields and enantiopurities when using 2-(ferrocenyl)ethanol or tryptophol. Such functional group tolerance encouraged us to apply the desymmetrization reaction of **1a** to the late-stage modification of structurally diverse alcohols derived from natural or bioactive molecules (Fig. 2C), including indomethacin, proline, biotin, and chenodeoxycholic acid, or bioactive alcohols (sulfurol, citronellol, protected glucose derivative). These desymmetrization reactions resulted in good-to-high yields of esters, with high levels of enantiopurity of the final product. For instance, the steroidal product **3r** and **3s** were obtained in high yields (67 and 66%) as single diastereomers (both 20:1 *dr*). In the reaction to steroidal product **3s**, the starting material contained three unprotected hydroxy groups. In this case, differences in the reaction rates of desymmetrization of secondary alcohols resulted in regioselectivity. Moreover, thiols also worked as esterification agents in this desymmetrization reaction (Fig. 2D), but their efficiency, in terms of yield and optical purities of thioesters **3w** and **3x**, was lower than that of the aforementioned esters.

To assess our method (Fig. 3), we introduced another prochiral [2.2]paracyclophane, namely *pseudo-gem*-diformylparacyclophane (**1b**). We began by optimizing the reaction conditions (for more details, please refer to Supplementary Table 10). After lowering the reaction temperature, we noted that the expected product **5a** was formed in excellent yield and enantiomeric purity (91%, 99.5/0.5 *er*) without the diester byproduct. In turn, by using the opposite enantiomeric form (*ent*-*pre*-**C1**), we gained access to the opposite enantiomer (*ent*-**5a**), obtaining the expected product in excellent yield and stereochemical outcomes. With sterically hindered alcohols, the reaction rate decreased, as expected, albeit without significantly affecting the enantiocontrol. Moreover, introducing different alcohols improved the yield and stereocontrol of the desymmetrization process.

## Mechanistic studies

To elucidate the reaction mechanism and origin of stereocontrol, we conducted control experiments with both substrates **1** (Figs. 4 and 5). First, treating **1a** (*pseudo-para*) with deuterated methanol-*d4* (Fig. 4A) under optimized conditions provided **3a**-*d3* with deuterated aldehyde (~40%, validated by ²H NMR), indicating the reversible formation of the Breslow intermediate. Subsequently, we studied the parallel kinetic isotope effect (Fig. 4B) using **1a** and **1a**-*d2* in a desymmetrization reaction with methanol under optimized reaction conditions for 1 h. The results showed a KIE (kinetic isotope effect) value of 2.8, implying that proton transfer in the formation of the Breslow intermediate is the rate-limiting step. To investigate the origin of enantiocontrol, we

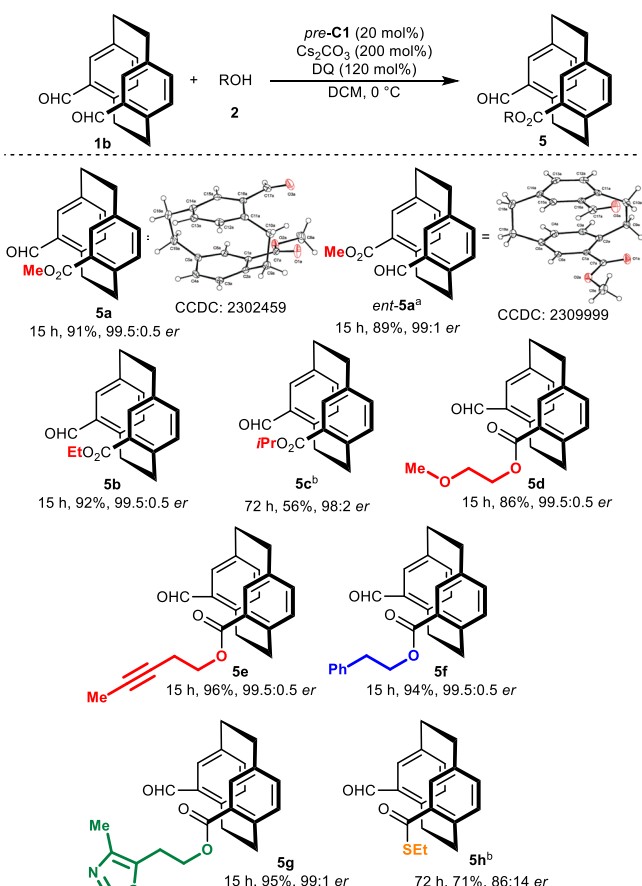

**Fig. 3 | Reaction scope of the *pseudo-gem* derivative.** Scope of aliphatic alcohols (highlighted in red). Scope of 2-arylethanols and related aromatic alcohols (highlighted in blue). Scope of alcohols derived from natural or bioactive compounds (highlighted in green). Scope of thiols (highlighted in orange).

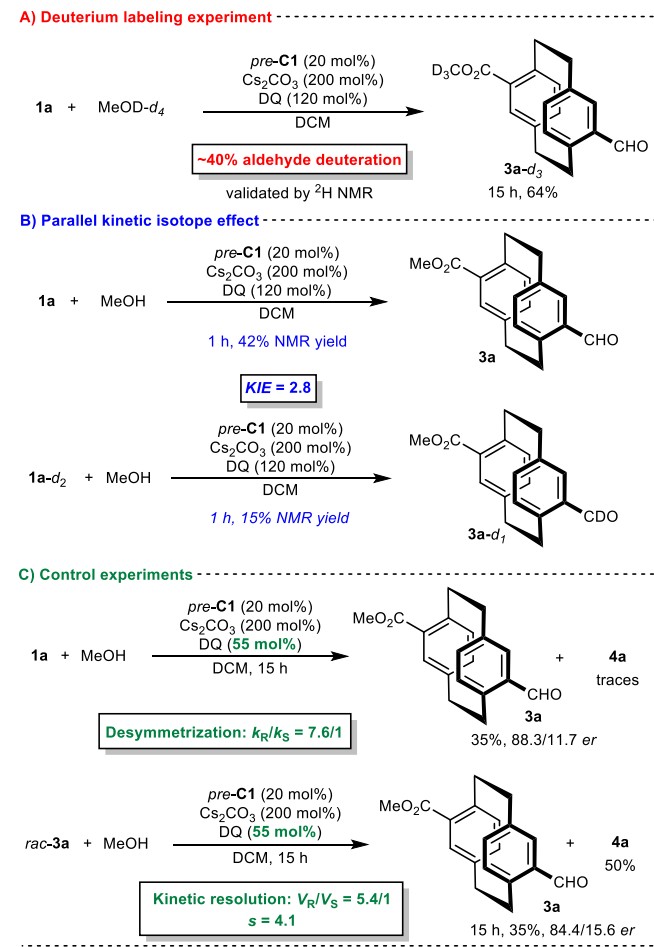

**Fig. 4 | Mechanistic studies for the *pseudo-para* derivative. A** Deuterium labeling experiment (highlighted in red). **B** Parallel kinetic isotope effect (highlighted in blue). **C** Control experiments (highlighted in green).

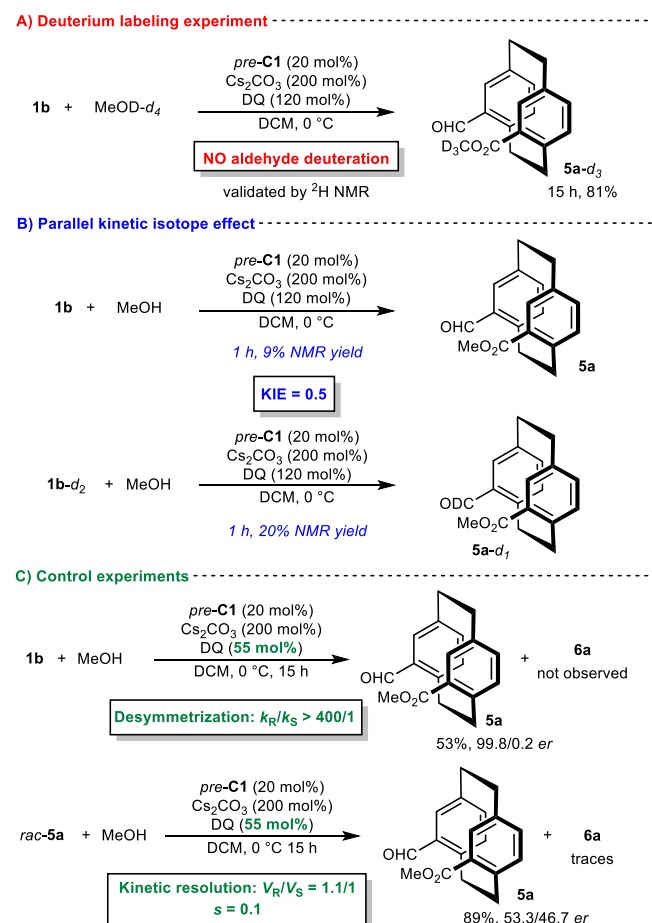

**Fig. 5 | Mechanistic studies for the *pseudo-gem* derivative. A** Deuterium labeling experiment (highlighted in red). **B** Parallel kinetic isotope effect (highlighted in blue). **C** Control experiments (highlighted in green).

conducted a series of control experiments (Fig. 4C). The model reaction with a lowered amount of oxidant (55 mol%) produced **3a** in 88:12 *er* with traces of the diesterification product, suggesting that desymmetrization is an enantiodivergent process and that kinetic resolution could be an additional enantiocontrol mechanism. To confirm this hypothesis, we conducted a kinetic resolution reaction of *rac*-**3a** under optimized reaction conditions with a lowered amount of oxidant (55 mol%), thereby forming enantioenriched product **3a** and confirming the existence of an additional source of enantiocontrol. Based on our findings, we propose that **1a** enantioselective desymmetrization ($k_R/k_S = 7.6/1$) is followed by kinetic resolution ($s = 4.1$), resulting in a high level of enantiocontrol (for details, please refer to pages 28-42 of the Supplementary Information file), in line with the slightly decreased enantiocontrol in the preparation of *ent*-**3a**.

We also performed another series of control experiments involving the desymmetrization of *pseudo-gem* derivative **1b** (Fig. 5). We noticed striking differences from the desymmetrization of **1a**. For example, we did not detect deuterium incorporation in the control reaction conducted with methanol-$d_4$ (Fig. 5A), indicating that the formation of the Breslow intermediate is an irreversible process. In the desymmetrization of **1b**, the KIE was significantly lower (~0.5). Accordingly, the initial carbene nucleophilic attack of **1b** is most likely the rate-limiting step (Fig. 5B). The origin of enantiocontrol was clear (Fig. 5C) because we observed nearly enantiopure product formation (99.8:0.2 *er*) in a control reaction of **1b** with a lowered amount of oxidant (55 mol%) under optimized conditions. Additionally, the kinetic resolution of *rac*-**5a** was ineffective ($s = 0.1$), indicating that

enantioselective desymmetrization ($k_R/k_S > 400/1$) is crucial for enantiocontrol in this process (for details, please refer to pages 28-42 of the Supplementary Information file). Based on these findings, *pseudo-gem*[2.2]paracyclophanes, not limited to dialdehydes, stand out as candidates for further elaboration in desymmetrization processes.

## Synthetic utilization of the chiral product

To showcase the practicality of this method, we performed a gram-scale desymmetrization of **1b** under optimized conditions (Fig. 6A). This gram-scale reaction provided us with access to a highly enantioenriched product **5a** in a high yield of 88% with 99.5/0.5 *er*. The follow-up reactions of the planar chiral product **5a** highlighted the usefulness and modulation of the aldehydic group (Fig. 6B). Moreover, the thioesterification reaction of **5a** promoted by oxidative NHC catalysis produced thioester **7** in excellent yield and retaining optical purity. Similarly, the corresponding olefin **8** was isolated in nearly quantitative yield as a product of the Wittig reaction. Through reductive methods, such as reductive amination or aldehyde reduction, we prepared secondary amine **9** and benzylic alcohol **10** in good-to-high yields, without significant changes in stereochemical outcomes. In addition, **11**, a crucial enantioenriched intermediate for preparing a valuable photocatalyst[56], was isolated by Pinnick oxidation in high yield and retaining optical purity.

Based on these results, we synthesized novel bifunctional catalysts by transforming both carbonyl groups (Fig. 6C). To prepare these novel catalysts, we began by conducting Bayer-Villiger oxidation followed by reduction and oxidation steps[57]. These steps yielded product

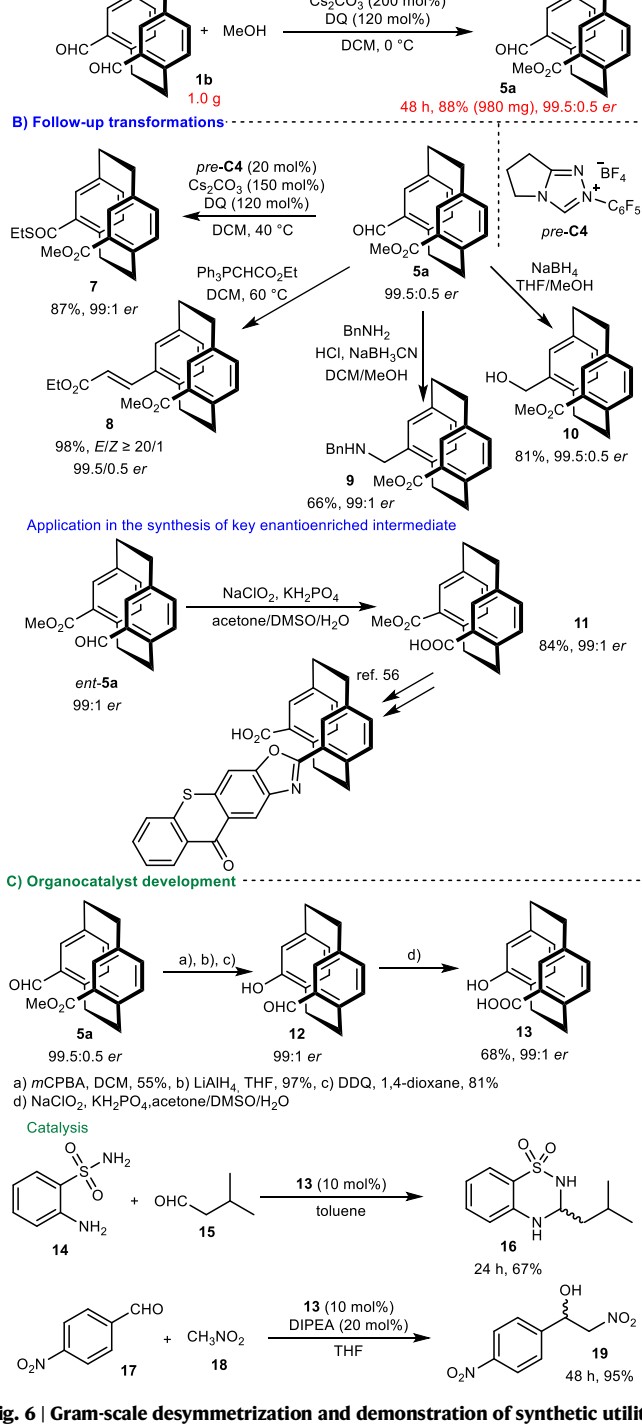

**Fig. 6 | Gram-scale desymmetrization and demonstration of synthetic utility.**
**A** Gram-scale reaction (highlighted in red). **B** Follow-up transformations (highlighted in blue). **C** Organocatalyst development (highlighted in green).

In summary[63], our metal-free methodology for NHC-catalyzed enantioselective desymmetrization of diformyl[2.2]paracyclophanes provides efficient access to highly enantioenriched planar chiral compounds. This operationally simple and effective strategy has a wide reaction scope with various alcohols involving natural and bioactive compounds. Moreover, the feasibility of the gram-scale desymmetrization reaction and the potential for diverse follow-up transformations underscore the value of this method. And as shown in our comprehensive experimental mechanistic studies, differences in the origin of enantiocontrol of *pseudo-para* and *pseudo-gem* diformyl derivatives in NHC-catalysed desymmetrizations identified *pseudo-gem* diformyl[2.2]paracyclophanes as valuable synthons for future elaborations. Accordingly, ongoing research into the synthesis of planar chiral molecules organocatalytic reactions and their applications in organocatalysis or novel ligand synthesis will continue in our laboratories.

## Methods

### Representative procedure

The vial (4 ml) was charged with **1** (26.4 mg, 0.1 mmol), *pre*-**C1** (7.8 mg, 0.02 mmol), DQ (49.0 mg, 0.12 mmol), and $Cs_2CO_3$ (65.2 mg, 0.2 mmol), followed by DCM (1.0 ml), and the corresponding alcohol (0.5 mmol) at the corresponding temperature. At this temperature, the reaction mixture was stirred for the indicated time. Once the reaction was completed by thin-layer chromatography (TLC), the solvent was evaporated. The crude product was purified by column chromatography (eluting by hexane/EtOAc mixtures).

## Data availability

The authors declare that the data supporting the findings of this study are available within the article and the Supplementary infomation file. The primary NMR FID files generated in this study have been deposited in the figshare repository (https://doi.org/10.6084/m9.figshare.24851235)[64]. Other detailed data are available from the corresponding authors upon request. The X-ray crystallographic coordinates for structures reported in this study have been deposited at the Cambridge Crystallographic Data Centre (CCDC), under deposition numbers 2302458, 2302459, and 2309999. These data can be obtained free of charge from The Cambridge Crystallographic Data Centre via www.ccdc.cam.ac.uk/data_request/cif.

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

## Acknowledgements

The authors gratefully acknowledge the Czech Science Foundation (24-12575 S, J.V.) for financial support. The authors also thank Dr. Štícha and Dr. Urban (both from Charles University) for the MS and IR analysis. Furthermore, the authors thank Dr. Carlos V. Melo (Charles University) for editing of the manuscript, and Dr. Tomáš Slanina (IOCB Prague) for discussing our mechanistic studies.

## Author contributions

V.D. designed project and performed the synthesis of all compounds. F. K. performed selected NMR experiments. I. C. performed X-ray analysis. J. V. conceived the study and directed the project. V.D., and J.V. wrote the manuscript. All authors have approved the final version of the manuscript.

## Competing interests

The authors declare no competing interests.
