## [Peer Review File · Nature Communications]

Organocatalytic Desymmetrization Provides Access to Planar Chiral [2.2]ParacyclophanesREVIEWER COMMENTS

Reviewer #1 (Remarks to the Author):

The authors disclosed in this manuscript the NHC-catalyzed desymmetrization reactions of two meso-[2.2]paracyclophane-based dicarbaldehydes. A diversity of alcohols could be adopted in these asymmetric esterification reactions, with the corresponding ester products afforded in moderate to good yields and stereoselectivities. Interestingly, two distinct reaction mechanisms were observed with the two different meso-dicarbaldehydes, although the reaction conditions were similar. The pseudo-para substrate underwent a reversible NHC addition and Breslow Intermediate formation, with a kinetic resolution process followed to achieve the enantioselectivities. The pseudo-ipso substrate went through a completely different pathway, with the NHC addition reaction selectively controlled by the chiralities of the substrate and the NHC catalyst. This is an interesting phenomenon, although more information needs to be provided. In addition, the authors also showed synthetic application of the afforded planar chiral products, which further added value to the current work. In all, I do believe this is an interesting piece of work and will attract the interest of a broad range of readers.

Therefore, I can support it to be published in Nat. Commun., although a very similar piece of work from our group had been directly rejected by the same journal several months ago without a chance for peer review. Nevertheless, modifications and additional information should be provided to further strengthen this work:

- 1) Introduction section: "Organocatalytic approaches are excellent tools such as amino acids and alkaloids." I suggest shrink or delete these words, since I do not find direct / close relationships of these expressions to the current work, and, in addition, many organic catalysts such as some trivalent phosphines are not stable.
- 2) Introduction section: "Consequently, asymmetric organocatalysis such as [2.2]paracyclophane derivatives." This is not correct, since several works have already appeared online with organocatalytic approaches to access chiral [2.2]paracyclophane derivatives. I have read a closely related work in Nat. Commun.(2023, 14, 5239), which I believe should be cited here.
- 3) Page 3, Line 62: "but only one study has reported such an approach thus far....." I do not believe this was correct... At least the above work in Nat. Commun. should be mentioned.
- 4) Line 71, "NHC precursors easily accessible from....." I suggest delete "easily", since it is not exactly correct: many NHCs are difficult to prepare.
- 5) Line 73, ref. 46 is strange. It is not suitable to cite a review here. Typical examples include but not limited to "Angew Chem Int Ed, 2016, 55: 1820–1824; Org Lett, 2019, 21: 6169–6172; Angew Chem Int Ed, 2022, 61: e202117340; Chin J Org Chem, 2022, 42: 2504–2514" should be cited here. In addition, I do not think these works are "central-to-axial" transfer. These are direct atroposelective esterifications.
- 6) Line 82, I do not think the current work involves "central-to-planar chirality transfer process". The current work is a direct asymmetric esterification. By the way, the "central-to-planar chirality transfer" should be deleted. I suggest "N-Heterocyclic Carbene-Catalyzed Desymmetric Esterification for access to Planar Chiral [2.2]Paracyclophanes".
- 7) For the reaction mechanism, I suggest to move more information from the SI to the main text. For instance, the process for the determination of k_R/k_S and V_R/V_S in Scheme 3c, and their function should be indicated.
- 8) The authors described the control experiment, analyzed the results and proposed the reaction process (desymmetrization and KR for different substrates). However, the reason that caused this difference should also be explained. Is it resulted from intramolecular H-bonding, or some steric hindrance? Please provide a complete postulated catalytic

mechanism clarifying the stereoselective inductions with the two reaction substrates clearly described.

9) Close related reviews on the NHC-catalyzed KR and desymmetrizations should be cited: *Synlett* 2013, 24, 1165-1169; *Chem. Asian J.* 2018, 13, 2149-2163; *Chem. Asian J.* 2018, 13, 2184-2194; *Synthesis* 2019, 51, 1871-1891; *Sci. China Chem.* 2023, 66, <https://doi.org/10.1007/s11426-022-1657-0>.

Reviewer #2 (Remarks to the Author):

This work reports a novel organocatalytic desymmetrization of diformyl[2,2]paracyclophanes, using NHC catalysts under oxidative conditions in the presence of various alcohols. This method delivers in a single step enantioenriched bifunctional paracyclophanes bearing both an aldehyde and an ester functionality, with a chemical diversity introduced on the ester group (22 examples). Thioesters can also be obtained, albeit with lower efficiency in terms of yield and optical purities.

Two different substrates have been used in this study: a centrosymmetric pseudo-para diformyl[2,2]paracyclophane, and a pseudo-gem (not pseudo-ipso) diformyl[2,2]paracyclophane exhibiting a plane of symmetry (not a center).

For the first substrate, yields range from 59 to 93% and e.r. are between 82:18 and 99:1 with primary alcohols. The reaction is less efficient with secondary alcohols, and is not working with *t*BuOH or phenols.

Best results are obtained with the pseudo-gem substrate, with e.r. generally better than 99:1 and yields between 56 and 96%.

Although the desymmetrization of centrosymmetric pseudo-para diformyl[2,2]paracyclophane has already been reported using asymmetric transfer hydrogenation ((ref. 43 cited in the paper), the desymmetrization of pseudo-gem substrate is a groundbreaking novelty since it was reported to give Cannizzaro type reactions under reductive conditions (ref. 43).

Furthermore, the authors conducted a very elegant and convincing mechanistic investigation to explain the observed stereocontrol of the reaction. It appears that the high enantioselectivity obtained from the pseudo-para bis-aldehyde is the result of an enantioconvergent process that combines desymmetrization and kinetic resolution, with a reversible formation of the Breslow intermediate, whereas the perfect enantioselectivity obtained from the pseudo-gem substrate is the result of a non-reversible enantioselective formation of Breslow intermediate.

All in all, this new methodology enables in a very simple manner access to very useful chiral building blocks. It will clearly impact the field of chiral paracyclophane chemistry.

The experimental part is very well described, with many additional details on parameter optimizations and all the products are well characterized. Several crystal structures secure the relative and absolute configuration determination.

I recommend publication, after a few improvements of the manuscript.

- centrosymmetry: compound 1b is NOT centrosymmetric! It has a plane of symmetry. This has to be corrected in all the manuscript.
- nomenclature: the description of substituted paracyclophanes has been proposed by Hopf. According to his pioneering work, compound 1b should be described as a "pseudo-gem" compound and not a "pseudo-ipso" compound.
- "substrate scope". I do agree that the scope of the reaction is broad if one considers the ester point of diversity, but the scope of the "substrate" is not, since only 2 different

compounds are desymmetrized in this work. I suggest to modify the "substrate scope " (Page 7, line 125) for a "reaction scope", and to remove the discussion on the so-called "restricted substrate scope" (page 3, line55) of ref 43.

- kinetic resolution. The maximum product yield of a kinetic resolution can be greater than 50%, but with e.e. lower than 100% Page 3, line 61, the sentence has to be modified "the maximum enantiopure product yield is only 50%)

- green process (page 5, lines 104-105). I do agree that electrooxydation is a green process, but here, 2 equiv. of TBAI are needed, and the side product of the reaction is probably HI. I'm not sure that this protocol is "greener" than the use of DQ, even if TBAI is called a "promoter" by the authors of this method.

- The use of unprotected sterols leads to the chemoselective esterification of primary alcohol. The selectivity is explained by the different rates of reaction between primary and secondary alcohols. However, the reaction took 48h, instead of 15h with lauryl alcohol. Could the final selectivity be the result of transesterification of transient secondary alcohols, leading to the formation of a single isomer?

- On the non-reversibility of the formation of Breslow intermediate for pseudo-gem substrate. Is it possible that alcoholate II (supporting info, page S26) reacts with the proximal aldehyde to form in a reversible manner an hemiacetal ? This competitive pathway could explain the difference between the two mechanisms. Could the authors design an experiment to monitor/trap this putative intermediate? May be using one equivalent of NCH, and no oxidant nor alcohol.

- Supporting information:

The numbering of the pre-catalysts is not the same as in the manuscript.

Ref 2: Two papers are cited, only the first one refers to the preparation of compounds 1a and 1b.

L. Micouin

Reviewer #3 (Remarks to the Author):

This paper describes an efficient and versatile protocol for organocatalytic desymmetrization esterification of centrosymmetric diformyl[2.2]paracyclophanes through NHC catalysis under mild conditions. The work is very meaningful for organic chemists. So I recommend this paper be published on this journal after the following points are considered.

1. In the line of 46, "4 carboxy[2.2]paracyclophane" should be written as "4-carboxy[2.2]paracyclophane".

2. In the line of 151, "we introduced another centrosymmetric [2.2]paracyclophane, namely pseudo-ipso-diformylparacyclophane" is misstated, pseudo-ipso-diformylparacyclophane is plane-symmetric.

3. In the Scheme 3: Mechanistic studies, control experiments (Scheme3C, left), "The model reaction with a lowered amount of oxidant (55 mol%) produced 3a in 88:12 er with traces of the diesterification product", but in the scheme, "Desymmetrization: $k_R/k_S = 5.4/1$ " might be wrong.

4. In the line of 213, "This operationally simple and effective strategy has a wide substrate scope" is a little exaggerated. For the [2.2]paracyclophane, only two substrates were introduced.

5. For the synthetic utility, there are many reactions involving the aldehyde-group, how about reductive amination, Wittig reaction and so on.

6. As for the pseudo-para [2.2]paracyclophane, the optimized reaction condition is at room

temperature, how about at 0°C? When at 0°C, would the enantiopurity of 3f and 3h be improved? And at 0°C, would the reaction mechanisms for both [2.2]paracyclophanes be same?

7. The synthetic utility of this method should be fully demonstrated. At least one of the examples listed in Figure 1 should be synthesized using this method.

Professor Jan Veselý
Charles University, Faculty of Science,
Department of Organic Chemistry
Hlavova 2030, 128 43 Praha 2, Czech Republic
Phone: +420-221-951-305, fax: 420 221 951 326,
E-mail: jxvesely@natur.cuni.cz

Dear reviewers,

Thank you for your detailed and thoughtful critiques. We have addressed all comments and modified the manuscript and the supplementary information accordingly. Below, please find our responses to your comments.

Reviewer #1

Comments:

1) Introduction section: "Organocatalytic approaches are excellent tools such as amino acids and alkaloids." I suggest shrink or delete these words, since I do not find direct / close relationships of these expressions to the current work, and, in addition, many organic catalysts such as some trivalent phosphines are not stable.

We thank the reviewer for the suggestion to revise this section for accuracy. As suggested, we have edited the statement on organocatalysis as follows: "Organocatalytic approaches are **valuable** tools for preparing enantiomerically pure compounds given the operational simplicity of their reactions, which frequently include water and air tolerance. In addition, **commonly used** organocatalysts are ~~particularly stable, diverse and~~ available in both enantiomeric forms, and often derived from natural sources, such as amino acids and alkaloids.", in line with reference 5 by Han et al. published in *Chem. Soc. Rev.* **50**, 1522-1586 (2021). These statements not only enhance the accuracy of this background information but also help us better frame the research question, further underscoring the limitations of this area of research and, therefore, the impact of our findings. For these reasons, we sincerely thank the reviewer for the input.

2) Introduction section: "Consequently, asymmetric organocatalysis such as [2.2]paracyclophane derivatives." This is not correct, since several works have already appeared online with organocatalytic approaches to access chiral [2.2]paracyclophane derivatives. I have read a closely related work in *Nat. Commun.* (2023, 14, 5239), which I believe should be cited here.

We thank the reviewer for the recommendation to mention the aforementioned study (*Nat. Commun* 2023) here in this section of the Introduction. However, this study had already been mentioned in the manuscript (ref 39 in the original draft of the manuscript). Moreover, we have carefully checked the literature, and we did not find any other work related to this topic. Therefore, our concluding sentence "Consequently, asymmetric organocatalysis applications remain overlooked, especially in the production of planar chiral molecules, such as [2.2]paracyclophane derivatives" is accurate; we do not state that no other study has been previously published on asymmetric organocatalysis applications but rather that such applications remain overlooked. In other words, our literature review confirms the veracity of our statement.

3) Page 3, Line 62: "but only one study has reported such an approach thus far....." I do not believe this was correct... At least the above work in *Nat. Commun.* should be mentioned.

As explained in the previous response, the aforementioned study (*Nat. Commun* 2023) is cited in the current version of the manuscript as reference 39.

4) Line 71, "NHC precursors easily accessible from....." I suggest delete "easily", since it is not exactly correct: many NHCs are difficult to prepare.

Faculty of Science
Charles University
Albertov 6, 128 43 Praha 2
www.natur.cuni.cz

Professor Jan Veselý
Charles University, Faculty of Science,
Department of Organic Chemistry
Hlavova 2030, 128 43 Praha 2, Czech Republic
Phone: +420-221-951-305, fax: 420 221 951 326,
E-mail: jxvesely@natur.cuni.cz

Once again, we thank the reviewer for carefully reading our manuscript and for the efforts to ensure the accuracy of all statements and word choice. As suggested, we have deleted “easily” in the revised manuscript.

5) Line 73, ref. 46 is strange. It is not suitable to cite a review here. Typical examples include but not limited to “Angew Chem Int Ed, 2016, 55: 1820–1824; Org Lett, 2019, 21: 6169–6172; Angew Chem Int Ed, 2022, 61: e202117340; Chin J Org Chem, 2022, 42: 2504–2514” should be cited here. In addition, I do not think these works are “central-to-axial” transfer. These are direct atroposelective esterifications.

We sincerely thank the reviewer for such a thorough review of our manuscript. We fully agree with the reviewer's opinion, and we have modified the manuscript accordingly, i.e. we have moved the reference related to NHC-catalyzed atroposelective esterification of aromatic aldehydes to this sentence (now ref. 52, previously ref. 46 in the original version of the manuscript) and cited the study suggested by the reviewer (Chin et al., *Chin. J. Org. Chem.* **42**, 2504-2514 (2022)) here, currently reference 53. We believe that we now cite all relevant studies in the current version of the manuscript because the other studies mentioned by the reviewer (two published in ACIE and one in Org Letters) are not directly related to the mentioned topic. So, after a carefully analysis of the reviewer's suggestion, we decided not to cite them.

As for the term used to describe the reactions, we have replaced “central-to-axial” by “atroposelective”. The corresponding sentence now reads: “For example, oxidative NHC catalysis was applied to the **atroposelective** desymmetrization of aromatic dialdehyde, producing axially chiral monoesters”. Once again, we thank the reviewer for enhancing the accuracy of our manuscript.

6) Line 82, I do not think the current work involves “central-to-planar chirality transfer process”. The current work is a direct asymmetric esterification. By the way, the “central-to-planar chirality transfer” should be deleted. I suggest “N-Heterocyclic Carbene-Catalyzed Desymmetric Esterification for access to Planar Chiral [2.2]Paracyclophanes”.

We thank the reviewer for the suggestion, and in line with our response to the previous comment (5), we have modified the title of the manuscript as follows: “**Organocatalytic Desymmetrization Provides Access to Planar Chiral [2.2]Paracyclophanes**”.

7) For the reaction mechanism, I suggest to move more information from the SI to the main text. For instance, the process for the determination of kR/kS and VR/VS in Scheme 3c, and their function should be indicated.

We thank the reviewer for this suggestion. We have added key information about enantioselective desymmetrization and kinetic resolutions of both types of [2.2]paracyclophanes to the revised manuscript. In addition, we have added the calculations with equations to the Supporting Information file.

8) The authors described the control experiment, analyzed the results and proposed the reaction process (desymmetrization and KR for different substrates). However, the reason that caused this difference should also be explained. Is it resulted from intramolecular H-bonding, or some steric hindrance? Please provide a complete postulated catalytic mechanism clarifying the stereoselective inductions with the two reaction substrates clearly described.

We thank the reviewer for this suggestion. Heeding the reviewer's advice, we have proposed putative transition states. We have added those states, together with an explanation to the revised Supplementary Information file (Schemes S2 and S4 together with the text). We now propose that the main enantiodiscrimination step is determined by steric hindrance of the chiral catalyst during an initial nucleophilic attack to the aldehyde group of the starting material. Bulkiness of the isopropyl group of the catalyst effectively

Faculty of Science
Charles University
Albertov 6, 128 43 Praha 2
www.natur.cuni.cz

Professor Jan Veselý
Charles University, Faculty of Science,
Department of Organic Chemistry
Hlavova 2030, 128 43 Praha 2, Czech Republic
Phone: +420-221-951-305, fax: 420 221 951 326,
E-mail: jxvesely@natur.cuni.cz

shields one of its faces. The less-shielded face is non-planar due to the presence of one of the methyl groups, which sterically disfavors the formation of one enantiomer.

9) Close related reviews on the NHC-catalyzed KR and desymmetrizations should be cited: Synlett 2013, 24, 1165-1169; Chem. Asian J. 2018, 13, 2149-2163; Chem. Asian J. 2018, 13, 2184-2194; Synthesis 2019, 51, 1871-1891; Sci. China Chem. 2023, 66, <https://doi.org/10.1007/s11426-022-1657-0>.

We thank the reviewer for these literature suggestions. We have included two of them in the revised version of our manuscript as references 48 and 50. In turn, Chem. Asian J. 2018, 13, 2149-2163 had already been mentioned in the original draft of our manuscript, previously as ref. 49 and now as ref. 51.

Reviewer #2 (L. Micouin)

Comments:

1) I recommend publication, after a few improvements of the manuscript.
- centrosymmetry: compound 1b is NOT centrosymmetric! It has a plane of symmetry. This has to be corrected in all the manuscript.

We would like to thank the reviewer for careful reading our manuscript. We fully agree with the reviewer's statement, and we have revised the manuscript accordingly, replacing 'centrosymmetric' for 'prochiral' throughout the text.

2) Nomenclature: the description of substituted paracyclophanes has been proposed by Hopf. According to his pioneering work, compound 1b should be described as a "pseudo-gem" compound and not a "pseudo-ipso" compound.

We apologize for this nomenclature inaccuracy. As suggested by the reviewer, we revised the manuscript and the supplementary file, replacing the term '*pseudo-ipso*' for '*pseudo-gem*'.

3) Substrate scope: I do agree that the scope of the reaction is broad if one considers the ester point of diversity, but the scope of the "substrate" is not, since only 2 different compounds are desymmetrized in this work. I suggest to modify the "substrate scope" (Page 7, line 125) for a "reaction scope", and to remove the discussion on the so-called "restricted substrate scope" (page 3, line55) of ref 43.

We thank the reviewer for the suggestion. We have modified the manuscript accordingly, replacing the term "substrate scope" for "reaction scope" throughout the text, where appropriate. In addition, we have replaced the term "restricted substrate scope" by "limited reaction scope". These changes enhance the accuracy of our statements.

4) Kinetic resolution. The maximum product yield of a kinetic resolution can be greater than 50%, but with e.e. lower than 100% Page 3, line 61, the sentence has to be modified "the maximum enantiopure product yield is only 50%)

We apologize for the inaccuracy, and we fully agree with the proposed change. As suggested, we have modified the corresponding sentence, which now reads "Regardless of the approach, though, kinetic resolution entails an inherent limitation, that is, the maximum **enantiopure** product yield is only 50%".

Professor Jan Veselý
Charles University, Faculty of Science,
Department of Organic Chemistry
Hlavova 2030, 128 43 Praha 2, Czech Republic
Phone: +420-221-951-305, fax: 420 221 951 326,
E-mail: jxvesely@natur.cuni.cz

5) Green process (page 5, lines 104-105). I do agree that electrooxydation is a green process, but here, 2 equiv. of TBAI are needed, and the side product of the reaction is probably HI. I'm not sure that this protocol is "greener" than the use of DQ, even if TBAI is called a "promoter" by the authors of this method.

We thank the reviewer for the insightful comment, and we understand and respect the reviewer's point of view. We have modified the text as suggested, i.e., we deleted the term 'greener'. The corresponding sentence now reads "Nevertheless, this experiment validated electrochemical oxidation as a potentially **more suitable** approach than other systems involving additional oxidants". Furthermore, in line with the title of the study (currently ref. 55) cited in the previous sentence and based on the referee's comments, we have replaced "TBAI" by "iodine as a promoter". The corresponding sentence now reads "Conversely, electroredox oxidation using **iodide as a promoter**⁵⁵ produced the expected product **3a** in 47% yield, albeit slightly decreasing the enantiocontrol (entry 7)"

6) The use of unprotected sterols leads to the chemoselective esterification of primary alcohol. The selectivity is explained by the different rates of reaction between primary and secondary alcohols. However, the reaction took 48h, instead of 15h with lauryl alcohol. Could the final selectivity be the result of transesterification of transient secondary alcohols, leading to the formation of a single isomer?

To test the reviewer's hypothesis, we have conducted an additional desymmetrization reaction of **1a** with chenodeoxycholanol protected at both secondary hydroxyl groups, *bis*-OMOM-chenodeoxycholanol. The reaction produced **3s** as a single diastereomer (20:1 *dr*) in high yield (66%) after 48h. These results are now included in the revised version of the manuscript (Scheme1) and in the Supplementary Information file. This observation is in line with the results of **1a** desymmetrization with an unprotected sterol. Based on these results, chemoselectivity likely derives from differences in the reaction rates of hindered alcohols rather than from transesterification of transient secondary alcohols.

7) On the non-reversibility of the formation of Breslow intermediate for pseudo-gem substrate. Is it possible that alcoholate II (supporting info, page S26) reacts with the proximal aldehyde to form in a reversible manner an hemiacetal? This competitive pathway could explain the difference between the two mechanisms. Could the authors design an experiment to monitor/trap this putative intermediate? May be using one equivalent of NCH, and no oxidant nor alcohol.

We sincerely thank the reviewer for such an excellent question based on which we designed two NMR experiments. In the first experiment, we attempted quantitative deprotonation of the carbene precursor with a strong base, such as DBU (in an equimolar amount), followed by addition of an equimolar amount of diformylparacyclophane **1b** in deuterated DCM at room temperature. In the second experiment, we tried to mimic optimized reaction conditions by using cesium carbonate instead of DBU. In the experiment with DBU, we were not able to detect any free carbene due to the formation of a complex mixture of products 5 minutes after adding paracyclophane derivative **1b**. Unfortunately, we did not observe any reaction, and aldehyde **1b** was identified without any changes after 120 minutes. In the experiment with cesium carbonate, we did not observe any reaction, including the formation of free carbene or any structural change of diformyl derivative **1b**, as shown below in the stacked ¹H spectra of both NMR experiments. So, we were not able to confirm (or disprove) the hypothesis suggested by the reviewer. We believe that the formation of free carbene is driven by its consumption in the following steps under optimized reaction conditions. Based on our literature search, we hypothesize that the 1,2-proton shift, which seems to be a rate-determining step, may be assisted by bases, especially carbonates. For further information on a related study, please refer to *Chem. Asian. J.* **16**, 2346-2350, (2021), <https://doi.org/10.1002/asia.202100351>.

Professor Jan Veselý
Charles University, Faculty of Science,
Department of Organic Chemistry
Hlavova 2030, 128 43 Praha 2, Czech Republic
Phone: +420-221-951-305, fax: 420 221 951 326,
E-mail: jxvesely@natur.cuni.cz

Professor Jan Veselý
Charles University, Faculty of Science,
Department of Organic Chemistry
Hlavova 2030, 128 43 Praha 2, Czech Republic
Phone: +420-221-951-305, fax: 420 221 951 326,
E-mail: jxvesely@natur.cuni.cz

8) Supporting information: The numbering of the pre-catalysts is not the same as in the manuscript.
Ref 2: Two papers are cited, only the first one refers to the preparation of compounds 1a and 1b.

We would like to thank the reviewer for carefully reading not only the manuscript but also the Supplementary Information. We apologize for all the aforementioned inaccuracies. Based on the reviewer's feedback, we updated the file by renumbering the precatalyst and removing the highlighted paper.

Reviewer #3

Comments:

1) In the line of 46, "4 carboxy[2.2]paracyclophane" should be written as "4-carboxy[2.2]paracyclophane".

We thank the reviewer for flagging this issue. We have corrected the term of the compound in the revised version of our manuscript by introducing the hyphen, as highlighted by the reviewer.

2) In the line of 151, "we introduced another centrosymmetric [2.2]paracyclophane, namely pseudo-ipsodiformylparacyclophane" is misstated, pseudo-ipso-diformylparacyclophane is plane-symmetric.

We fully agree with the reviewer's comment, and we have revised the manuscript accordingly by replacing the term 'centrosymmetric' by 'prochiral'.

3) In the Scheme 3: Mechanistic studies, control experiments (Scheme3C, left), "The model reaction with a lowered amount of oxidant (55 mol%) produced 3a in 88:12 er with traces of the diesterification product", but in the scheme, "Desymmetrization: $k_R/k_S = 5.4/1$ " might be wrong.

We apologize for the error in our calculations, which were revealed upon careful review. We have fixed the errors and updated the incorrect values in Scheme 3 of the revised version of our manuscript. In fact, prompted by this error, we have revised all our calculations and identified a numerical mistake in the calculation of the selectivity factor. To ensure accuracy, we have thoroughly cross-checked all values and provided further details on our calculations in the revised Supplementary Information file.

4) In the line of 213, "This operationally simple and effective strategy has a wide substrate scope" is a little exaggerated. For the [2.2]paracyclophane, only two substrates were introduced.

We thank the reviewer for the comment, and we have modified the corresponding sentence as follows: "This operationally simple and effective strategy has a wide **reaction** scope with various alcohols involving natural and bioactive compounds."

5) For the synthetic utility, there are many reactions involving the aldehyde-group, how about reductive amination, Wittig reaction and so on.

Based on the reviewer's suggestion, we performed several follow-up transformations to demonstrate the synthetic utility of the aldehyde group. We performed a series of transformations such as NHC-catalyzed thioesterification, Wittig reaction, reductive amination, and reduction. In all examples, we identified the corresponding products in high-to-excellent yields without significant changes in optical purities. These experiments are discussed in the main text and Scheme 4B of the revised version of our manuscript. The

Faculty of Science
Charles University
Albertov 6, 128 43 Praha 2
www.natur.cuni.cz

Professor Jan Veselý
Charles University, Faculty of Science,
Department of Organic Chemistry
Hlavova 2030, 128 43 Praha 2, Czech Republic
Phone: +420-221-951-305, fax: 420 221 951 326,
E-mail: jxvesely@natur.cuni.cz

corresponding experimental procedures and spectroscopic data were added to the current Supplementary Information file.

6) As for the pseudo-para [2.2]paracyclophane, the optimized reaction condition is at room temperature, how about at 0°C? When at 0°C, would the enantiopurity of **3f** and **3h** be improved? And at 0°C, would the reaction mechanisms for both [2.2]paracyclophanes be same?

We sincerely thank the reviewer for raising these pertinent questions. To answer them, we set up a model reaction of **1a** under optimized reaction conditions using methanol at 0 °C. Unfortunately, we did not observe any significant formation of the desired product. This result was added to the revised version of the Supplementary Information file (Table S8, Entry 7). Additionally, we performed a desymmetrization reaction of **1a** at 0°C with suggested alcohols, with similar results (only traces of product **3f** were detected after 72 hours) in the reaction with 2-bromoethanol. However, we observed full conversion after 72 hours in desymmetrization reaction leading to **3h** using pent-4-en-1-ol as a starting alcohol. We isolated the corresponding product in 44% yield with excellent optical purity (99:1 *er*). This yield is significantly lower due to the formation of a diester. Based on these results, we identified kinetic resolution as the main stereocontrol process. To validate this hypothesis, we performed a control reaction using 55 mol% of DQ under an inert atmosphere. After 24 hours, we isolated the corresponding product **3h** in 29% yield and 96:4 *er* without a significant amount of diester product. Accordingly, the original hypothesis about the origin of stereocontrol remains valid.

To better understand both desymmetrization processes, we performed further control experiments at room temperature from pseudo-geminal derivative (**1b**) instead of **1a** at 0°C, which proved ineffective. As reported in the original draft of the manuscript, we performed another series of control experiments. However, we did not observe any aldehyde deuteration in the desymmetrization reaction conducted with deuterated methanol (the corresponding product (**5a-d₃**) was isolated in 86% yield). Our study of the parallel kinetic isotopic effect revealed that KIE = 1.2. Notably, the NMR yield was higher than 50% after 1 hour for both products. Moreover, the origin of stereocontrol was in line with our previous experiments at 0°C. With a lower amount of DQ (55 mol%), we observed the formation of the desired product after 15 hours in 37% with 98.4/1.6 *er* ($k_R/k_S = 62/1$). After 15 hours of reaction at room temperature, kinetic resolution of rac-**5a** provided 62% of recovered product **5a** in 64.1/35.9 *er* ($V_R/V_S = 1.8/1$, $s = 1.4$). Based on these results, all conclusions regarding the mechanism of origin of stereocontrol remain valid.

7) The synthetic utility of this method should be fully demonstrated. At least one of the examples listed in Figure 1 should be synthesized using this method.

To demonstrate that our synthesis method is straightforward and to avoid preparing two similar types of organocatalysts, we changed the example of catalyst (Fig. 1C) in the revised manuscript. Product **5a** (accessible in both enantiomeric forms in the desymmetrization process) was successfully transformed into the corresponding acid **11** under Pinick oxidation conditions in high yield, without losing optical purity. Acid **11** is the direct enantioenriched catalyst precursor shown in Fig. 1 of the revised version of our manuscript. This synthesis protocol has been previously described in the literature (*Org. Biomol. Chem.* **21**, 9330-9336 (2023), <https://doi.org/10.1039/D3OB01580G>). This example highlights the value of our desymmetrization approach, and follow-up transformation, given the complicated access to enantioenriched intermediate **11**, as reported in the aforementioned study.

Sincerely Yours

Jan Veselý
Faculty of Science
Charles University
Albertov 6, 128 43 Praha 2
www.natur.cuni.cz

REVIEWERS' COMMENTS

Reviewer #1 (Remarks to the Author):

The authors have addressed all my concerns. Now the manuscript is more precisely presented and is fit to Nat. Commun.. I do not have additional suggestions.

Reviewer #2 (Remarks to the Author):

The authors responded to all the points raised in my first review and corrected the manuscript accordingly. There may still be a minor point to be checked in the SI: the absolute values of the optical rotations between compound 5a and its enantiomer are very different. It is possible that the value given for compound 5a is not the correct one.

This work deserves publication.

Reviewer #3 (Remarks to the Author):

The authors had revised the manuscript according the reviewers' comments and I suggested that the manuscript can be accepted now.

Professor Jan Veselý
Charles University, Faculty of Science,
Department of Organic Chemistry
Hlavova 2030, 128 43 Praha 2, Czech Republic
Phone: +420-221-951-305, fax: 420 221 951 326,
E-mail: jxvesely@natur.cuni.cz

Dear reviewers,

Thank you for your detailed review process and positive feedback for the revised manuscript, currently entitled: Organocatalytic Desymmetrization Provides Access to Planar Chiral [2.2]Paracyclophanes (ID: NCOMMS-23-62156A). We have addressed additional comment from Reviewer #2 and modified the supplementary information accordingly. Below, please find our response.

Reviewer #2

There may still be a minor point to be checked in the SI: the absolute values of the optical rotations between compound 5a and its enantiomer are very different. It is possible that the value given for compound 5a is not the correct one.

We sincerely thank the reviewer for such a thorough review of our manuscript and Supplementary information. We remeasured the suggested optical rotations for both highlighted enantiomers at the same concentration and corrected both optical rotation values.

Sincerely Yours

Jan Veselý